# Establishment of a Uniaxial Tensile Fracture Inversion Model Based on Fracture Surface Reconstruction (FRASTA)

Haidong Jia [1], Zhenghao Jiao [2], Lianshuang Dai [3], Yongbin Que [1], Qingshan Feng [3], Ming Yang [1] and Yuguang Cao [2,*]

1    PipeChina West Pipeline Company, Urumqi 830013, China
2    College of Pipeline and Civil Engineering, China University of Petroleum (East China),
     Qingdao 266580, China
3    PipeChina Company, Beijing 100013, China
*    Correspondence: caoyuguang@upc.edu.cn

**Abstract:** In order to infer the load on the component after the experimental uniaxial tensile fracture inversion model based on cross-sectional reconstruction, (FRASTA) was proposed to infer the load on the tested components. This model can combine the fracture surface characteristics of experimental specimens to reconstruct the fracture surface morphology and invert the fracture process of uniaxial tensile specimens. Based on the assumption of rectangular rod fracture, a quantitative inversion model for a unidirectional stress load based on dissipative plasticity characteristics was established, and the inversion results based on cross-sectional reconstruction were compared with the experimental measurement results. The results indicate that when only considering the unidirectional stress state, the two have a high degree of consistency, with a maximum measurement error of 5.3%, fully verifying the accuracy of the fracture surface reconstruction and inversion model.

**Keywords:** fracture surface; fracture process; FRASTA; DIC

## 1. Introduction

The rapid development of oil and gas pipeline construction has made pipeline failure analysis increasingly important, with the failure analysis of pipeline girth-welded joints accounting for the majority of the analyses. Circumferential welded joints include base metal, heat-affected zone, and weld type. Because of the complexity of the materials, the research on the material properties becomes the primary research task. Chen [1] studied the tensile behavior of high-strength steel and its welding metals at room temperature to extremely low temperatures and proposed a temperature-dependent mechanical performance prediction equation. Basso [2] studied the tensile properties of austempered ductile iron with different ferrite contents and proposed a correlation between strength and elongation.

Traditional mechanical performance testing requires damage to the specimen in most cases and cannot evaluate in-service equipment. At present, micro-losses or non-destructive testing methods are used to determine certain parameters, and empirical or standard formulas are used to calculate the material's mechanical performance parameters. For example, the spherical indentation method can be used to test the yield strength and tensile strength of metal plate components [3]. Bozca [4] believed that there was a hidden functional relationship between material strength and hardness, and the bending strength under quasi-static load was evaluated by measuring the hardness distribution of the gear components. Li [5] used ultrasonic testing to measure the surface wave velocity of metal materials at different positions and angles and obtained the mechanical properties at the corresponding positions and angles. For special process specimens and materials, Raju [6] established a mechanical characterization method based on uniaxial tensile testing using statistical analysis.

Xu [7] pointed out that the history of the temperature and strain rates experienced by the material itself seriously affects the original mechanical properties measured through

standard tensile tests; that is, the testing of material properties is sensitive to the environment and needs to reduce the influence of other factors. At the same time, due to the heterogeneity of real materials, macroscopic mechanical performance testing gradually cannot meet the needs of real materials. Therefore, research has been conducted on the fracture behavior of materials at the micro-level. The fracture results after fracture are a good research object. Mandelbrot [8] pioneered the method of applying fractal geometry to the quantitative analysis of metal fractures. As a non-destructive method for preserving fracture surface specimens, a computer-aided stereo-matching method using scanning electron microscopy (stereo pairs) has been developed for reconstructing and analyzing three-dimensional fracture surfaces in materials [9]. Tanaka [10] proposed a fractal analysis method for characterizing the anisotropy of fracture surfaces in metal materials and compared the results of two-dimensional and three-dimensional fractal analyses. Invasive restructuring was carried out by Zhao for three-dimensional fractures in the same coordinate system, and the characteristics of various structures were analyzed [11]. Kobayashi studied the fatigue load spectrum of aluminum sheet fractures by reconstructing their three-dimensional morphology characteristics [12]. Ammann used image processing and stereological principles to restore the three-dimensional morphology of fractures from scanning electron microscopy images [13]. Abell studied the relationship between concrete fracture surface roughness and fracture using laser-focused microscopy and visual density technology in the laboratory of the University of Illinois in the United States [14]. Carlos used atomic force microscopy to study the microstructure and roughness of fracture surfaces of different plastic materials [15]. Zhou used a laser profilometer to measure the morphology of rock cross-sections and evaluate their reconstruction results and anisotropy [16]. Yamamoto reconstructed the entire process of fracture notch opening using scanning electron microscopy and demonstrated the crack propagation evolution of the cross-section [17]. Kuroda developed a three-dimensional morphology reconstruction system for stereo scanning electron microscopy parallax measurement, reconstructed the cleavage fracture surface of ferritic stainless steel, and measured the relative height of its river patterns [18]. Boccaccina obtained the relative height values of fracture surfaces of glass and a glass matrix with different volume fraction composite materials added using contour curve technology and studied the toughness and ductility of glass with composite materials added [19]. Lopez conducted a study on the relationship between the morphology characteristics of fracture reconstruction results and fracture mechanisms. Through a series of shear mechanics experiments, they established the relationship between mechanical parameters and microstructure morphology [20].

At present, the quantitative analysis of fractures is still in its infancy. This article proposes a fracture analysis method based on uniaxial tensile specimens, establishes a reconstruction and matching method for the scanning data characteristics of fractures in uniaxial tensile specimens, and proposes the assumption of a rectangular bar with fracture based on the unidirectional stress state of uniaxial tensile specimens, thereby forming a quantitative reverse inference model for a unidirectional stress load based on dissipative plastic properties. The load was quantitatively inferred from the microscopic fracture data.

## 2. Experiment

In this paper, the size of the uniaxial tensile specimen selected is shown in Figure 1, and the chemical composition of the test material is shown in Table 1. The experimental materials were provided by the National Pipeline Network Group. The experiment used positioning displacement loading, and during the experiment, the digital image correlation (DIC) method was used to obtain the engineering stress–strain curve of X80 pipeline steel. DIC is a new strain measurement technology that calculates the full-field strain on the surface of a specimen by calculating the position changes in the same pixel. The equipment we used was VIC-3D produced in Japan. DIC has the following advantages: Nnon-contact and non-destructive measurement: DIC can measure strain and displacement without physical contact, thereby maintaining the integrity of the sample for further

analysis. Full field measurement: Unlike traditional point by point measurement techniques, DIC provides a comprehensive view of the entire surface deformation field of the specimen, providing richer data for analysis. High spatial resolution and precision: The high resolution of DIC allows for the detection of small deformations, which helps its accuracy and reliability in capturing subtle differences in material behavior under load. The resolution of strain is higher than that of conventional contact extensometers, and it can measure the strain value at the necking point of uniaxial tensile specimens. We used the Shimadzu AGS-X testing machine produced in Japan. The uniaxial tensile test was conducted on the CTM9100 microcomputer-controlled electronic universal testing machine, with a strain rate of $0.005 \cdot s^{-1}$ during stretching. Before the start of the experiment, we first sprayed white matte paint with uniform and moderate thickness on the surface of the test piece as the base color and then used black matte paint for spraying. When spraying black matte paint, the nozzle should be at a certain distance from the specimen and there should be no wind around it to ensure that black paint particles can evenly fall on the surface of the white matte paint, forming evenly distributed black speckle. As a reference point for DIC measurement, the area of black speckle should account for about 50% of the total area. After the spraying is completed, the positions of the two charge-coupled device (CCD) cameras were calibrated using a 7 mm calibration board to establish the three-dimensional coordinates of the scattered spots on the surface of the specimen. After the experiment began, the specimen was loaded by a universal testing machine, while the DIC device recorded 10 images per second. The DIC testing device is shown in Figure 2. Two CCD cameras were used to record the stretching process, and the deformation process of the uniaxial tensile specimen surface at room temperature was obtained through VIC-3D software 3.0 processing. In order to ensure the accuracy of the test, three sets of parallel tests were conducted. The load of the entire test process was obtained through a tensile testing machine, and the deformation of the gauge segment was obtained through DIC equipment. The engineering stress of the material was obtained by dividing the load by the initial cross-sectional area of the uniaxial tensile specimen, and the engineering strain of the material was obtained by dividing the deformation by the length of the gauge segment. The load displacement curve at the time of specimen fracture is shown in Figure 3. By processing the stress–strain curve of material engineering according to ASTM standards, the elastic modulus $E$ = 206 GPa, yield strength $R_{P0.2}$ = 594 MPa, and tensile strength $R_m$ = 713 MPa of X80 pipeline steel could be obtained.

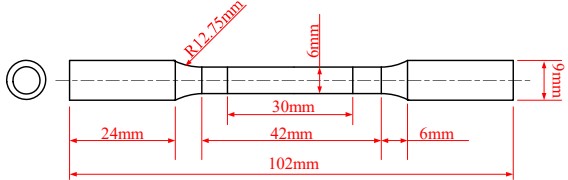

**Figure 1.** Design drawing of uniaxial tensile specimen.

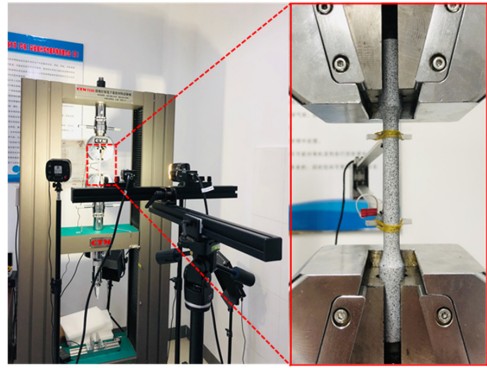

**Figure 2.** Schematic diagram of DIC shooting mode.

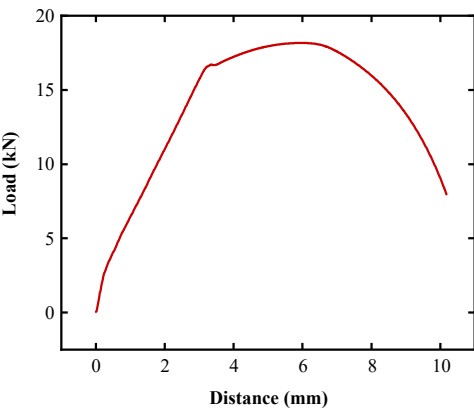

**Figure 3.** Load displacement curve measured experimentally.

**Table 1.** Main chemical composition of pipeline steel (mass fraction).

| C | Si | Mn | P | S | Cr | Mo | Ni | Nb | V | Ti | Cu |
|------|------|------|------|-------|-------|------|------|------|-------|-------|------|
| 0.04 | 0.23 | 1.87 | 0.01 | 0.003 | 0.025 | 0.27 | 0.23 | 0.06 | 0.006 | 0.017 | 0.13 |

## 3. Reconstruction of Fracture Surface and Inversion of Fracture Process

We used a laser scanner to scan the surface of the fracture and achieve reconstruction of the fracture surface. Subsequently, in order to better deduce the fracture process of the specimen, MATLAB was used to process the data, complete the three-dimensional reconstruction of the fracture surface, match the characteristics of the fracture surface, and invert the fracture process.

The relative height of the fracture surface was used to characterize the morphology of the fracture, and a KEYENCE laser scanner was used to scan the fracture of the three-point bending specimen after the experiment. Based on the shape and size of the uniaxial tensile specimen after fracture, test marks and stage positioning lines were selected for specimen positioning to ensure that the initial positions of the upper and lower fracture surfaces of the same specimen were the same, as shown in Figure 4.

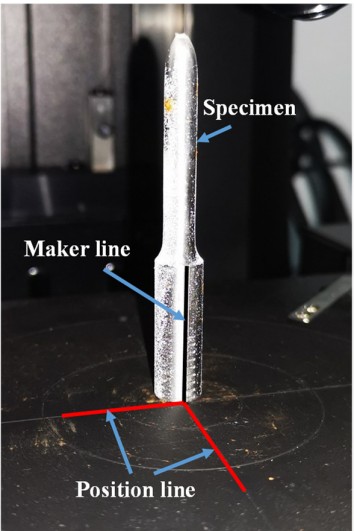

**Figure 4.** Specimen positioning diagram.

In order to ensure the scanning accuracy of the fracture surface, the fracture surface was divided into four equal parts and a 5× magnification objective was selected to scan the fracture area. Finally, a splicing measurement method was used to obtain the overall fracture result, with a scanning measurement accuracy of 0.001 μm. The total scanning

data of the fracture were $1024 \times 1448$ data points. Before inverting the fracture process, it was necessary to match the initial state of the specimen. Firstly, the macroscopic matching of the upper and lower fracture surfaces was carried out using the axis scratches on the surface of the specimen. Due to the limitations of the scanning instrument and the shape of the deformed specimen, it was not possible to record the entire deformation part of the specimen, as shown in Figure 5 (3D reconstruction of upper and lower fracture surfaces). The height of the part that can be recorded is about 2000. Simultaneously considering the shape of the uniaxial tensile specimen, the scanning results of uniaxial tensile were processed using MATLAB, as shown in Figure 6 (matching diagram of upper and lower fracture surfaces). The cross-sectional line obtained by cutting through the y-z plane of the selected specimen axis is shown in Figure 7 (matching diagram of upper and lower fracture section heights).

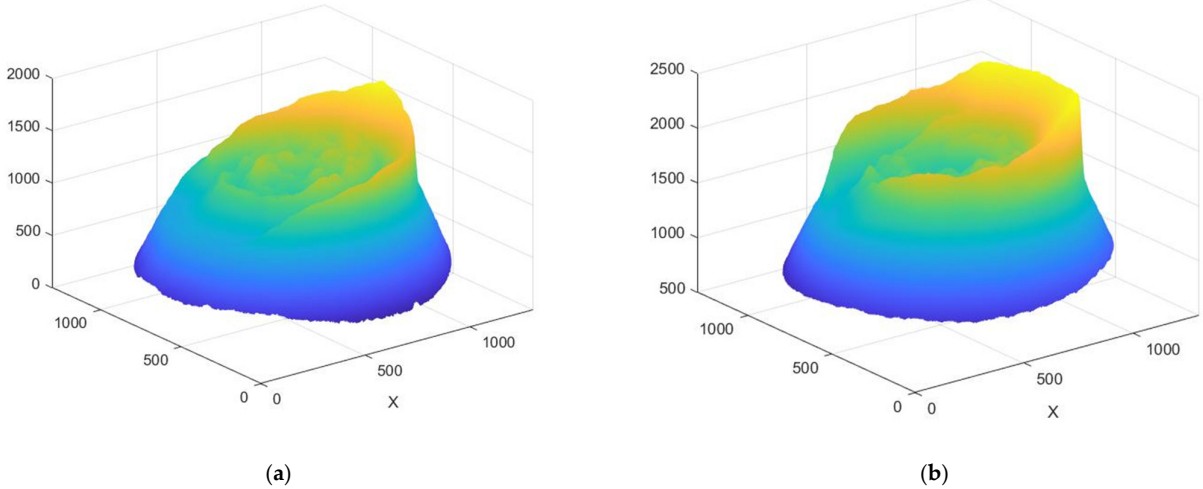

(**a**)                                                             (**b**)

**Figure 5.** Fracture scanning result graph. (**a**) Reconstruction of three-dimensional morphology of upper fracture surface. (**b**) Reconstruction of 3D morphology of lower fracture surface. (X: Length (mm)).

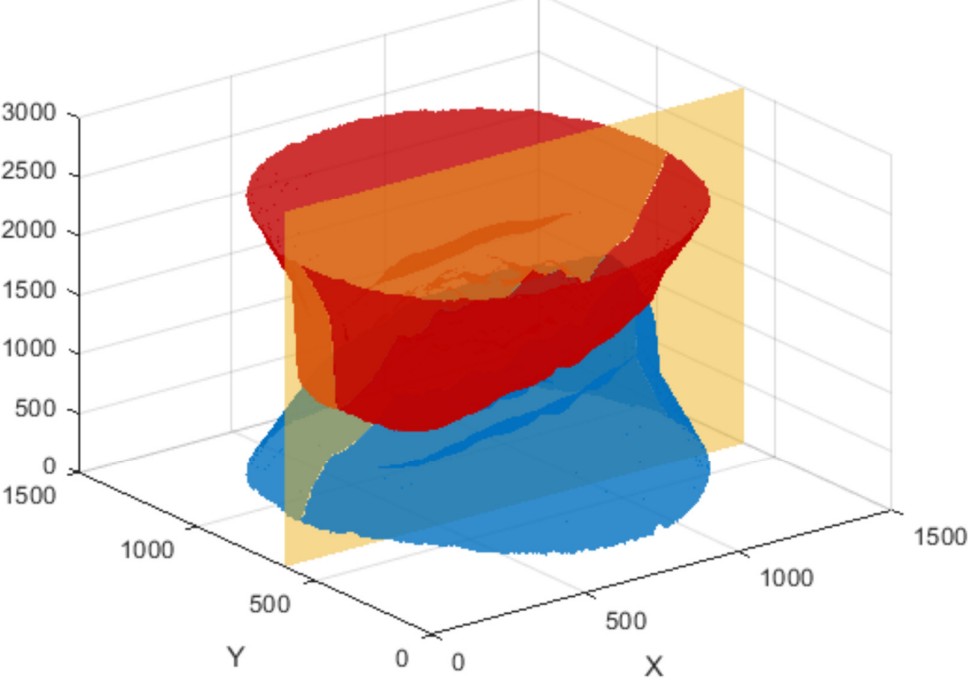

**Figure 6.** Initial state inversion results. (X: Length (mm) and Y: Width (mm)).

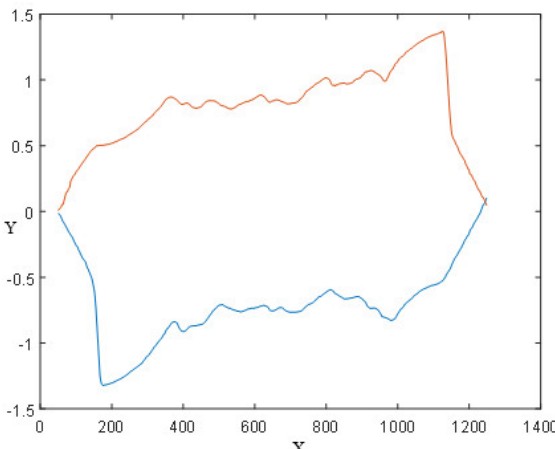

**Figure 7.** Lines of fracture surface (middle section). (X: Length (mm) and Y: Height (mm)).

Next, we fixed the lower fracture surface and rotated the upper fracture surface to find the best matching position. The difference between the upper and lower fracture surfaces of the reference position, with the non-rotating position as the reference, and the corresponding position is shown in Figure 8 (matching diagram of initial fracture interface curve). At a rotation scale of 0.5°, the difference between the corresponding positions of the upper and lower fracture surfaces after rotation is shown in Figure 9 and Table 2 (differences in height curves at different angles). It can be observed that the difference on the right side gradually increases after clockwise rotation and gradually increases after counterclockwise rotation on the left side.

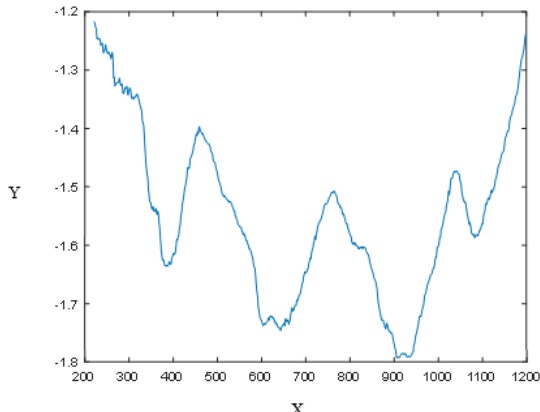

**Figure 8.** The difference between the upper and lower fracture surfaces of corresponding position (reference position). (X: Length (mm) and Y: Height (mm)).

**Table 2.** Result of fracture difference at different rotation angles.

| Rotation Angle | −1° | −0.5° | 0° | 0.5° | 1° |
|---|---|---|---|---|---|
| Difference range | 1.804 | 1.695 | 0.576 | 0.938 | 1.062 |

Due to the macroscopic displacement of the uniaxial tensile specimen only along the axis direction of the specimen, it is only necessary to use a fixed lower fracture and a translational upper fracture in the z-direction to invert the fracture process of the specimen. The inversion results are shown in Figure 10 (inversion and experimental charts at different stages). Based on the fracture displacement, the inversion results of the fracture process correspond to the experimental results. It can be found that the macroscopic deformation of the uniaxial tensile specimen is in good agreement, providing an accurate prerequisite for the subsequent quantitative inversion of external loads.

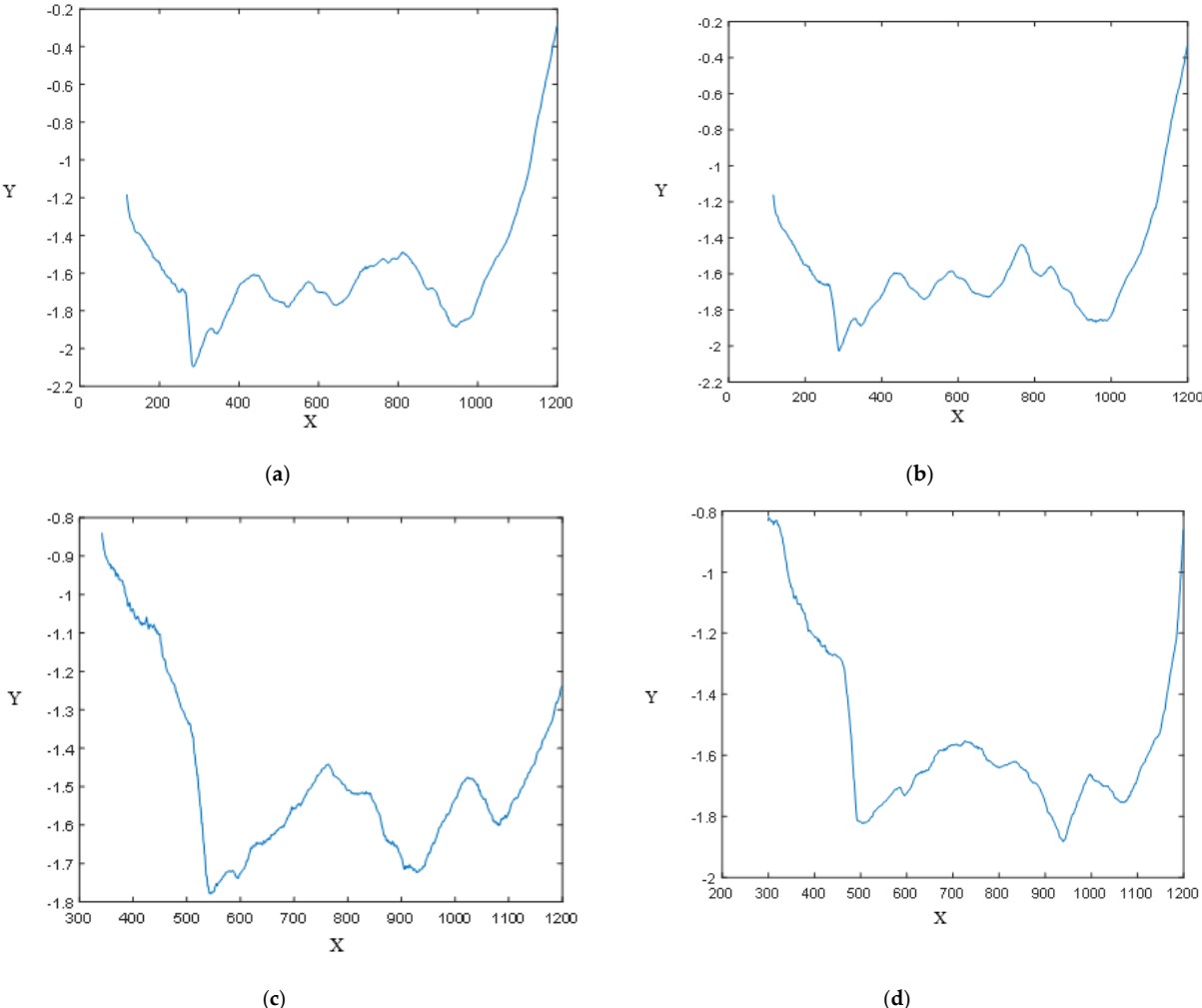

**Figure 9.** The difference between the upper and lower fracture surfaces of corresponding position (different rotation angles). (**a**) Diagram of the difference in height curve for rotation of 0.5° up and down. (**b**) Diagram of the difference in height curve for rotation of 1° up and down. (**c**) Diagram of the difference in height curve for rotation of 1.5° up and down. (**d**) Diagram of the difference in height curve for rotation of 2° up and down. (X: Length (mm) and Y: Height (mm)).

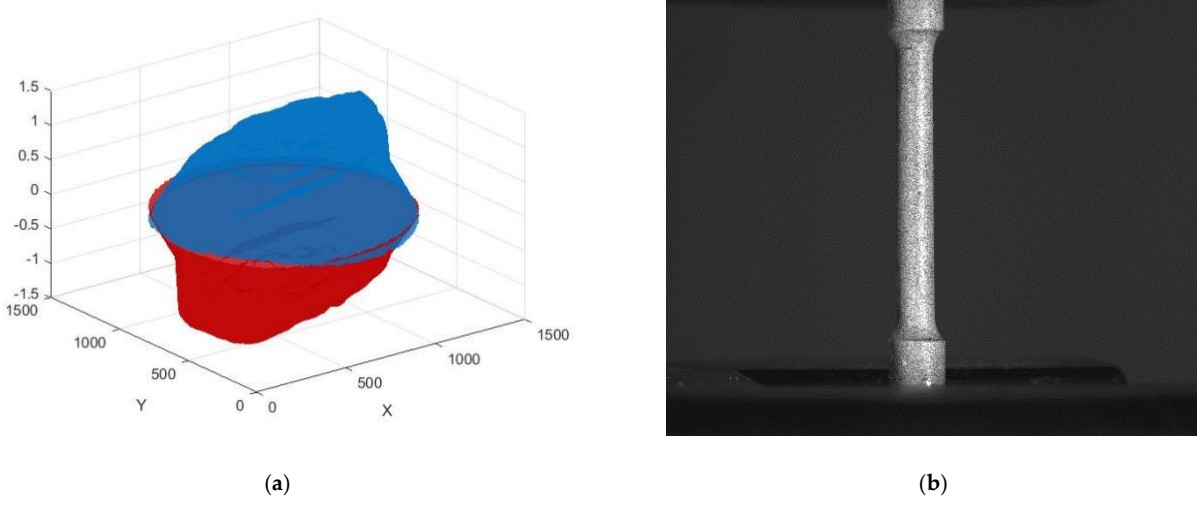

(**a**)  (**b**)

**Figure 10.** *Cont.*

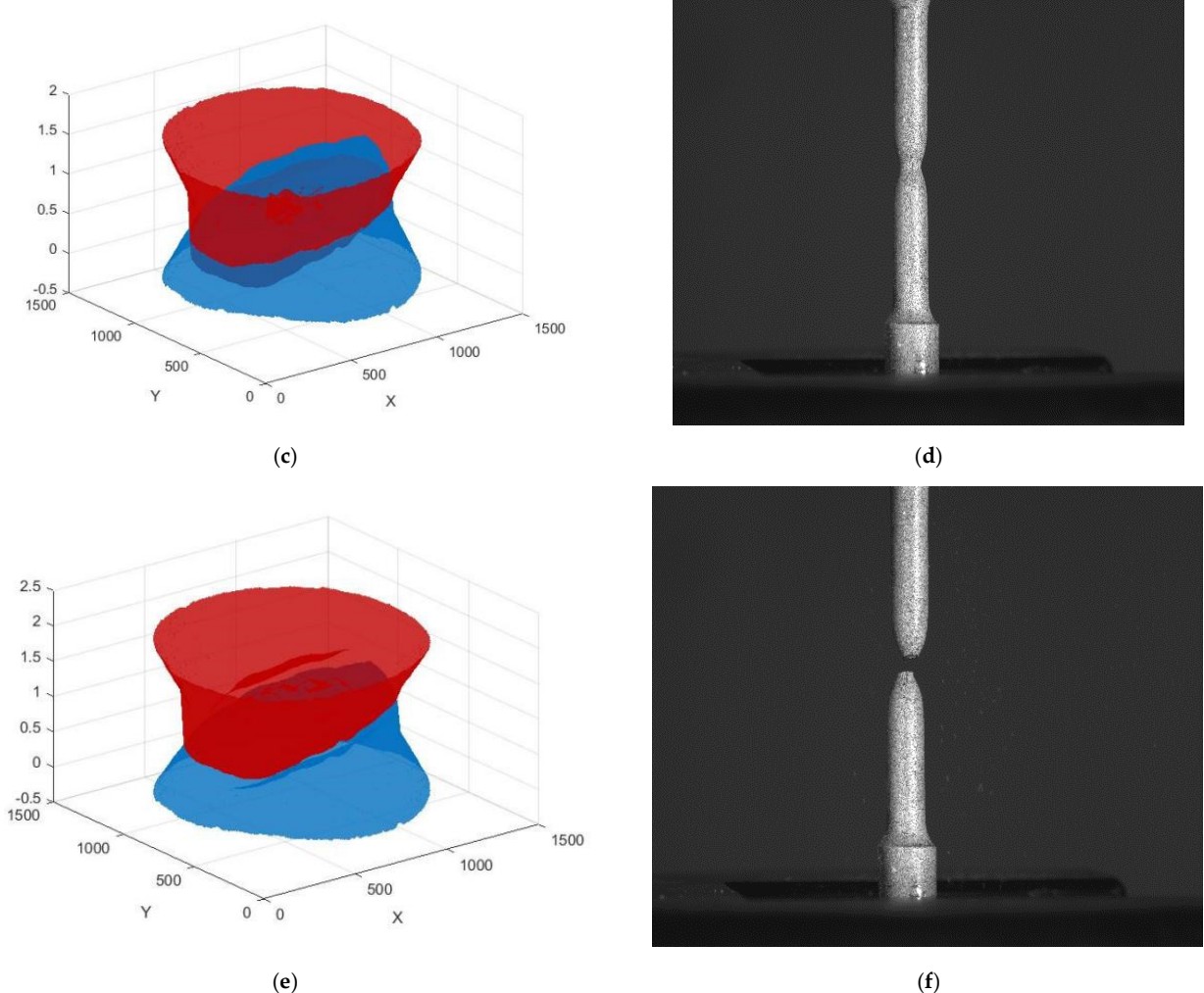

**Figure 10.** Inversion results and comparison of fracture processes. (**a**) Initial stage inversion. (**b**) Initial stage testing. (**c**) Enhancement stage inversion. (**d**) Strengthening phase testing. (**e**) Fault stage inversion. (**f**) Fracture stage test. (X: Length (mm) and Y: Width (mm)).

## 4. Load Measurement

Due to the scanning method of the laser scanner, the height value of the scanned fracture surface was recorded in a matrix form. Therefore, the original data recording form of the fracture surface of the uniaxial tensile specimen was to approximate the circular cross-section with a square cross-section, as shown in Figure 11. Based on the characteristics of the data and the morphology of the fracture surface, a uniaxial tensile fracture local tensile model based on the assumption of a rectangular rod is proposed. Assuming that the specimen consists of several small rectangular rods, the elongation deformation of the specimen can be discretized into the deformation of the rectangular rod. The fracture of all rectangular rods forms the macroscopic fracture of the specimen, and the fracture cross-section of each rectangular rod forms the uniaxial tensile fracture obtained after the test.

Considering that the high toughness steel of the pipeline is a stable material, there are

$$\Delta\sigma \cdot \Delta\varepsilon \geq 0 \tag{1}$$

It can be determined that the direction of stress increment is the same as that of strain increment. Therefore, the stress changes in the rectangular rod can be determined by the degree of deformation.

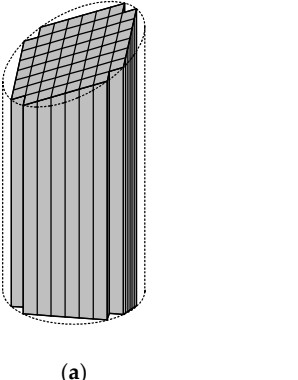

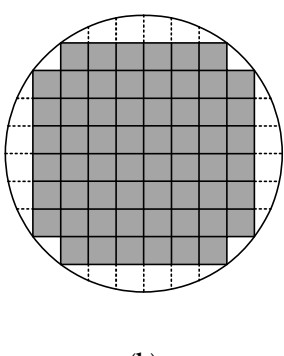

(a)　　　　　　　　　　　　　　　　(b)

**Figure 11.** Assumption of rectangular bars for uniaxial tensile specimens. (**a**) Three-dimensional view of rectangular pole. (**b**) Top view of rectangular pole.

The uniaxial tensile specimen is in a macroscopic state of unidirectional stress. Assuming that the rectangular rod remains in a unidirectional stress state throughout the deformation process, while ignoring the influence between the rectangular rods, it is only subjected to tensile stress throughout the entire experimental process, and the stress in the rectangular rod does not change axially. Figure 12 shows the deformation and stress–strain changes in a rectangular rod. The stress path is A–B–C, reaching the elastic limit of the material at point B and experiencing the maximum stress and strain before fracture at point C. At the moment of fracture, the state changes from C to D, leaving residual strain AD in the material. The area ABCD represents the dissipated energy of the material after stretching and fracture processes.

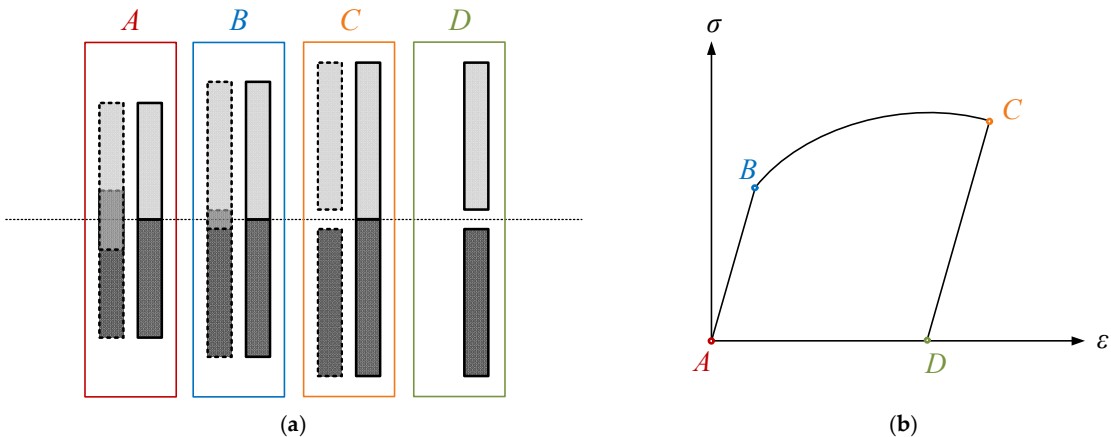

(a)　　　　　　　　　　　　　　　　(b)

**Figure 12.** Deformation of rectangular rod. (**a**) Fracture situation of rectangular rod. (**b**) Changes in stress path.

The strain energy density in a rectangular bar is

$$v_e(z) = \int_0^{\varepsilon_f} \sigma(z)\mathrm{d}\varepsilon \tag{2}$$

where $v_e$ is the strain energy density; $\varepsilon_f$ is the engineering fracture strain. The energy $u$ required for any rectangular rod to break is

$$u = \int_h v_e s\, \mathrm{d}h \tag{3}$$

where $s$ is the cross-sectional area of the rectangular rod and $h$ is the height of the rectangular rod, which can be obtained from fracture scanning data. Based on the local tensile model of the fracture surface, it is assumed that the plastic properties dissipated by the fracture of a

uniaxial tensile specimen are the sum of the plastic properties dissipated by the fracture of a rectangular rod during crack propagation, which is

$$U = \sum_N u \tag{4}$$

The fracture energy of the specimen is completely provided by external loads, as shown in Figure 13. When the displacement of the specimen increases from $V_{n-1}$ to $V_n$, it can be obtained by the difference method:

$$F_n = \frac{2\Delta U}{\Delta V} - F_{n-1} = \frac{2(U_n - U_{n-1})}{V_n - V_{n-1}} - F_{n-1} \tag{5}$$

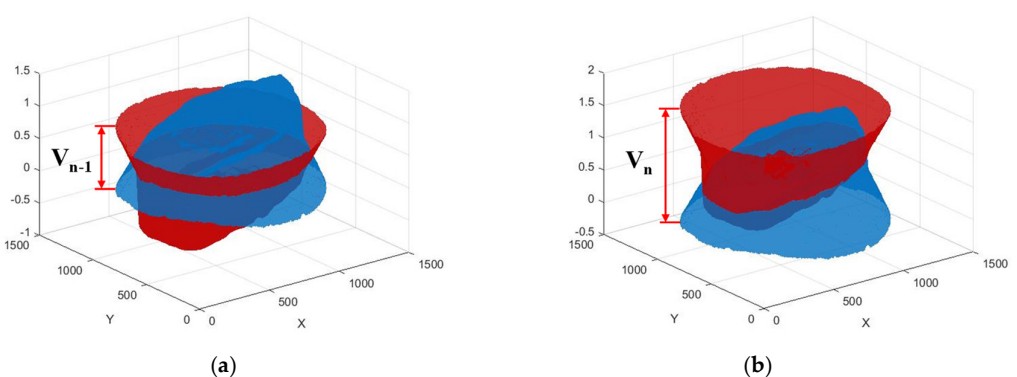

(**a**)　　　　　　　　　　　　　　　　　　(**b**)

**Figure 13.** Time difference of the specimen. (**a**) $V_{n-1}$ time. (**b**) $V_n$ time. (X: Length (mm) and Y: Width (mm)).

The comparison between the results of the load inversion model established based on the FRASTA method and the experimental measurement results is shown in Figure 14. It can be found that the FRASTA results are only slightly smaller than the experimental results, and the measurement curves of the two methods are almost the same, with a maximum measurement error of 5.3%, fully verifying the accuracy of the load inversion model.

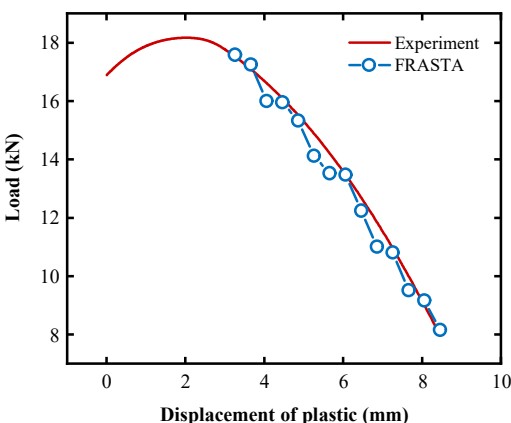

**Figure 14.** Comparison diagram of external load measurement results.

## 5. Conclusions

The fracture inversion model based on fracture surface reconstruction (FRASTA) method can effectively reconstruct the fracture process of uniaxial tensile specimens and has good correspondence with the load displacement points and DIC images recorded in experiments. It can visually analyze the changes in the sample at different times.

The assumption of a rectangular rod can simplify the stress–strain field of the specimen while retaining the microscopic characteristics of the fracture surface. By combining the

inversion model with the reconstruction of the fracture process, the external load of the specimen can be measured solely through fracture height data, which are highly consistent with the load displacement curve recorded in the experiment, fully verifying the accuracy of the inversion model based on fracture surface reconstruction.

**Author Contributions:** H.J.: Conceptualization and Investigation. Z.J.: Writing—Original Draft, Data curation, Methodology, and Experiment. L.D.: Investigation. Y.Q.: Investigation. Q.F.: Investigation. M.Y.: Investigation. Y.C.: Writing—Reviewing and Editing. All authors have read and agreed to the published version of the manuscript.

**Funding:** This paper was funded by the National Natural Science Foundation of China (12272412), National Natural Science Foundation of China (12202502), Shandong Provincial Natural Science Foundation, China (ZR2020ME093), Shandong Provincial Natural Science Foundation, China (ZR2022QE039), and Research on failure mechanism for girth weld of high steel pipeline (WZXGL202105).

**Institutional Review Board Statement:** Not applicable.

**Informed Consent Statement:** Not applicable.

**Data Availability Statement:** The original contributions presented in the study are included in the article, further inquiries can be directed to the corresponding author.

**Conflicts of Interest:** Lianshuang Dai and Qingshan Feng were employed by the PipeChina Company. Haidong Jia, Yongbin Qu and Ming Yang were employed by the PipeChina West Pipeline Company. The remaining authors declare that the research was conducted in the absence of any commercial or financial relationships that could be construed as a potential conflict of interest.

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
