# Peer review of "Establishment of a Uniaxial Tensile Fracture Inversion Model Based on Fracture Surface Reconstruction (FRASTA)"

_coatings, doi:10.3390/coatings14040489_

Round 1

Reviewer 1 Report

Comments and Suggestions for Authors

The paper is well-written and structured. The following minor revisions should be conducted by the authors before acceptance for publication.

1.     Fig. 6 is not clear. Please improve the quality of the figure.

2.     The numbers in Fig. 5 are not clear.

3.     What are the parameters and units in the X and Y axes of Figs. 5–10?

4.     The points A, B, C, D in Fig. 12 should be O, A, B, C. See the text in Lines 203–207.

5.     Please describe the mechanical behavior in Fig. 14. Why the load at zero displacement is 17 kN, the maximum load occurs at a displacement of 2 mm, and the load decreases with increasing the displacement further?

Comments on the Quality of English Language

Minor editing of English language required

Author Response

1. We have modified Figure 6
2. We have modified Figure 5
3. The parameters on the X-axis and Y-axis represent the size of the dimension, in millimeters
4. We have made modifications to the path written in the article to match Figure 12
5. We have selected the load displacement curve from the maximum load to the stage of complete fracture failure. Therefore, the load at zero displacement is 17kN, and the maximum load occurs at a displacement of 2mm. The load decreases as the displacement further increases

Reviewer 2 Report

Comments and Suggestions for Authors

The study of the properties of construction materials and the kinetics of their destruction is an important problem of today's engineering practice and the quantitative analysis of fractures is still in its infancy. The manuscript presents a crack inversion model based on crack surface reconstruction (FRASTA). The research is well designed but not very clearly presented. In order to achieve the goal set in the work, a detailed comparative analysis of existing publications related to research on the properties of materials and the peculiarities of their behavior under tensile conditions was carried out. The methodological section of the manuscript is laid out in sufficient detail. The authors used modern experimental equipment (the digital image correlation method, KEYENCE laser scanner, etc.) to test samples, as well as appropriate software to help interpret the results. They concluded that the method makes it possible to effectively reconstruct the process of destruction of uniaxial tensile specimens, and has good agreement with the load displacement points and DIC images recorded in the experiments.

However, some shortcomings should be corrected to make the manuscript acceptable for publication in Coatings.

1. Please state manufacturer, city and country from where equipment has been sourced. This have to be done for each equipment, software, material and chemical in the paper.

2. The manufacturer should be indicated when describing the material under study. Besides, the authors should give a reference where the data listed in Table 1 were taken from.

3. Since the basis of your experimental studies is the method of digital image correlation, I propose to describe in more detail the method of determining local deformations by this method and point out its advantages.

3. Eliminate the inconsistency of the markings “A-B-C-D-O” in Figure 12 and in the text.

4. If possible, the font sizes in Figures 6 - 10 should be increased.

5. Inscriptions in fig. 6 are not recognized. 
6. The authors should pay great attention to the use of words in singular/plural forms.

The authors presented a valuable scientific work. I recommend publishing this paper after correction of the shortcomings.

Author Response

Dear Reviewer:

Thank you very much for your valuable feedback and suggestions. Based on your suggestion, the following are the main revisions I have made to the paper:

1.We have explained the manufacturer, city, and country of origin of the equipment

2.The materials and their composition combinations are provided by the National Pipeline Group

3.We pointed out the advantages of DIC measurement in the article

4.We have made modifications to the content corresponding to Figure 12 to facilitate the matching of images and text

5.We have made the images clearer

6.The parameters on the X-axis and Y-axis represent the size of the dimension, in millimeters

7.We have made modifications to the use of words